# Spatially Offset Raman Spectroscopy Combined with Attention-Based LSTM for Freshness Evaluation of Shrimp

**DOI:** 10.3390/s23052827

**Published:** 2023-03-05

**Authors:** Zhenfang Liu, Yu Yang, Min Huang, Qibing Zhu

**Affiliations:** Key Laboratory of Advanced Process Control for Light Industry (Ministry of Education), Jiangnan University, Wuxi 214122, China

**Keywords:** spatially offset Raman spectroscopy, attention, LSTM, freshness evolution, shrimp

## Abstract

Optical detection of the freshness of intact in-shell shrimps is a well-known difficult task due to shell occlusion and its signal interference. The spatially offset Raman spectroscopy (SORS) is a workable technical solution for identifying and extracting subsurface shrimp meat information by collecting Raman scattering images at different distances from the offset laser incidence point. However, the SORS technology still suffers from physical information loss, difficulties in determining the optimum offset distance, and human operational errors. Thus, this paper presents a shrimp freshness detection method using spatially offset Raman spectroscopy combined with a targeted attention-based long short-term memory network (attention-based LSTM). The proposed attention-based LSTM model uses the LSTM module to extract physical and chemical composition information of tissue, weight the output of each module by an attention mechanism, and come together as a fully connected (FC) module for feature fusion and storage dates prediction. Modeling predictions by collecting Raman scattering images of 100 shrimps within 7 days. The *R*^2^, *RMSE*, and *RPD* of the attention-based LSTM model achieved 0.93, 0.48, and 4.06, respectively, which is superior to the conventional machine learning algorithm with manual selection of the optimal spatially offset distance. This method of automatically extracting information from SORS data by Attention-based LSTM eliminates human error and enables fast and non-destructive quality inspection of in-shell shrimp.

## 1. Introduction

Shrimp (*Fenneropenaeus chinensis*) have rich protein content, a reasonable ratio of amino acids required by the human body, delicious meat, and are becoming a popular fishery product. Meanwhile, due to their delicate muscle tissue and high autolysis enzyme activity, shrimp are highly susceptible to spoilage and deterioration during storage and transportation, resulting in food safety problems [1]. Therefore, rapid and non-destructive detection of shrimp quality is important to improve product quality and protect the rights and interests of consumers.

Traditional detection means of shrimp, such as physicochemical experiments, microbiology, gas–liquid phase [2], etc., have high detection accuracy, but also have problems such as lengthy time, waste of resources, and destruction of samples. Spectroscopic analysis techniques are used for the quality inspection of shrimp meat with the advantages of being rapid, being non-destructive, and having high accuracy [3]. The near-infrared and Raman spectroscopy detection techniques with molecular characterization have a unique ability to interpret the mechanism. For shrimp with high moisture content, water-insensitive Raman spectroscopy has higher detection accuracy [4]. However, the penetration depth of conventional backscattered Raman spectroscopy is limited, while shrimp are usually stored and sold with the shell intact. The thin and translucent shrimp shell blocks the Raman photon transmission, and the Raman spectrum of the shell also interferes with the freshness detection of the internal shrimp meat. There are no universal means of detecting internal food quality with surface interference, and the recently developed spatially offset Raman spectroscopy (SORS) provides a candidate with means for completely nondestructive detection.

The SORS technique is a novel Raman spectroscopy technique proposed by Matousek to overcome the measurement drawbacks of complex layered samples [5]. This technique offsets the laser incidence point and detection position, effectively suppressing the Raman signal and fluorescence interference of the surface material and obtaining the spectral information of the deep material two orders of magnitude deeper than the conventional Raman spectroscopy detection [6]. Qin et al. [7,8] developed a line-scan hyperspectral detection system to acquire SORS data, combined with optimal offset distance spectral selection to reduce surface signal interference, and applied it to quantify the lycopene content of intact tomatoes. Afseth et al. [9] demonstrated that the SORS technique can monitor the carotenoid content of intact salmon through the skin qualitatively and quantitatively. Liu et al. [10] detected the freshness of intact in-shell shrimps by the SORS technique, which quantifies the storage time of shrimp by choosing the optimal offset distance for Raman spectroscopy combined with a machine learning prediction model. These methods of selecting the optimal offset distance spectra for subsurface detection are susceptible to the influence of the optical properties, the surface layer thickness, and the instrument noise [11,12]. Meanwhile, the physical properties of shrimp meat change with the change in freshness, and the scattering profile at a specific waveband can reflect the physical properties of shrimp. Using Raman spectroscopy at a specific offset distance will lose spatial physical information, which will affect the construction of high-precision models. Finally, because of the band correlation of the Raman spectral data, the traditional machine learning model treats the Raman spectral data as an unordered vector, which may further lead to the loss of potentially useful information and bring potential risk of model over-fitting. Hence, in-depth detection of bilayer food samples is still a problem to be solved, and an inspection technique needs to be developed to solve the physical information loss of SORS tissue distribution, perceived errors in the selection of offset distances, and the redundancy of adjacent band information in Raman spectroscopy for subsurface detection of foods with surface interference (freshness of in-shell shrimp herein). With the development of deep learning and its application in the field of spectral detection, the deep network for processing sequence data represented by LSTM is widely acknowledged [13]. Therefore, based on existing research, this study proposes a targeted, non-destructive detection method for the freshness of in-shell shrimp using SORS technology combined with attention-based LSTM, which aims to:(1)Use a line-scan Raman hyperspectral imaging system to collect shrimp scattering images and perform pre-processing and spectral analysis;(2)Build the LSTM layer to extract the shrimp tissue scattering features and chemical composition spectral features, use the attention layer to weight the output of the LSTM module, and train the model;(3)Validate the advantages of the attention-based LSTM model in assessing the freshness of in-shell shrimp by comparing different structures of attention-based LSTM and machine learning models.

## 2. Materials and Methods

### 2.1. Sample Preparation

The experimental samples were from the same batch of 100 live shrimps bought from a supermarket (Auchan Investment Co., Ltd., Shanghai, China) in Wuxi, China. The weight of individual shrimp was 15 ± 1 g. Shrimp were blended in chopped ice over 20 min to indirect cadaverous death. Sterile paper towels were used to dry the surface of the shrimp. To simulate actual storage and marketing environments, shrimp were placed in a constant temperature and humidity refrigerator at four degrees Celsius. Shrimp were placed in fused quartz vessels, and Raman scattering images were acquired at intervals of 24 h. A total of 700 images were obtained by continuous collection for 7 days, and each image was treated as a sample.

### 2.2. Instrument and Experiment

Figure 1 shows the line-scan Raman hyperspectral imaging system based on a point source used for data acquisition in this study. The system mainly includes a 785 nm laser (I0785MM0500MF, Innovative Photonic Solutions, South Brunswick, NJ, USA) as the excitation source for Raman spectroscopy, an imaging spectrograph (ImSpector R10E, Specim, Oulu, Finland), and a CCD camera (iKon-M 934, Andor Technology, Hartford, CT, USA) to form the detection mechanism for Raman scattering images. The horizontal displacement platform is used for independent repetitive sampling of the sample at different positions. All components are integrated in a dark box to ensure that the detection process is not influenced by the external environment. The system was described in detail in the previous article [10].

In the experiment, the sample is placed on an electric displacement platform, and the height of the platform is adjusted to make the detection lens focus on the surface of the sample. The laser power was adjusted to 350 mW and the exposure time was selected to 5 s to ensure an intense Raman signal without destroying the physical and chemical properties of the sample. The camera window is 102.4 mm spatial distance, divided into 512 pixels. Platform movement is controlled by a motion controller and acquired ten times at 3 mm intervals. Raman scattering images of in-shell shrimp were acquired and saved at this parameter setting. Meanwhile, the reference spectra of shrimp meat and shell were the Raman spectra acquired at the laser point (backscattered Raman) with the same parameters.

### 2.3. Scattering Image Preprocessing

Raman scattering images from ten different positions form a three-dimensional image cube, which is derived as original data (Figure 2a). Each scattering image consists of *x* spatial positions and *λ* bands of spectral intensity. The 3D-cube data derived from this study contain not only valid information for model evaluation, but also interference information such as out-of-sample background images, invalid wavebands, outliers, and noise interference, which require data pre-processing. At first, a spatial range of 1 cm on the surface of the shrimp (5 mm to the left and right of the laser spot) was selected as the region of interest, and the wavelength range of 600–1800 cm^−1^ was used as the waveband of interest to streamline the data. An optical reflection abnormality region exists in the exoskeleton at the intersection of the thoracic carapace of the shrimp shell, and Raman spectra collected at this position are usually devoid of the typical Raman peaks of shrimp meat. Typical Raman spectral peaks in the reference spectra of shrimp meat were identified and the wavebands were recorded. Meanwhile, the typical spectral peaks of the spatially dimensional averaged spectra of the in-shell shrimp scattered images were also recorded, and the anomalous spectra of the shrimp shells corresponding to the scattered images were removed by comparison of the peak positions with the typical peaks (Figure 2b). The interference of cosmic rays usually produces abrupt narrow peaks, which were eliminated by removal of the maximum and minimum values in each waveband. Then the spectra independently and repeatedly collected from different locations on each shrimp were averaged and smoothed (Savitzky-Golay) to remove random noise caused by mechanical vibration (Figure 2c). Stokes lines were used to convert Raman wavebands into Raman shifts to reduce the interference of excitation wavebands [14].

Although LSTM has a long-term memory function, its memory capacity still cannot fully cover the whole Raman spectrum. Meanwhile, considering that Raman spectra have a strong correlation in adjacent wavebands, the adjacent waveband values are averaged to remove redundant information. The Raman scattering image is acquired by a straight line-scan centered on the laser incident point, and to reduce the effect of sample inhomogeneity, the images on both sides are flapped and averaged according to the laser point position (Figure 2d). Finally, a 200 × 11 matrix is selected for subsequent network processing, where 200 wavebands are used as temporal data and 11 offset positions are used as spatial feature vectors (Figure 2e). The Raman scattering profile for each waveband contains tissue distribution properties; thus, the profile is used as input to the network (Figure 2f), taking into account the spectral information between the wavebands.

### 2.4. Attention-Based LSTM Model

#### 2.4.1. Model Structure

Considering that the shrimp quality deterioration process affects both the molecular chemical structure and the physical tissue distribution, a deep learning model (Figure 3) named attention-based LSTM was built to extract effective information from Raman scattering images to realize shrimp freshness prediction, which mainly includes: (1) an input layer of scattering images; (2) an LSTM layer to extract scattering information and spectral information; (3) an attention layer to weight the output of LSTM module; and (4) an output layer to integrate features and regression prediction.

The input layer is the processed 2D Raman scattering image consisting of the profile (x1,x2,…,xm,  xi∈ℝ11,  m=200), where the 200 wavebands uniformly cover 600–1800 cm^−1^ and the 11 spatial positions uniformly cover the offset positions within 5 mm of the laser point. The LSTM layer is a feedback neural network structure dealing with sequence features as a modification of the recurrent neural network, which can solve the gradient disappearance problem in long-term memory and has been applied in Raman spectral analysis [15]. The LSTM carries out feature selection and storage through the interaction between four chained repetitive cells. Firstly, the redundant information in the former input temporary cell state (ht-1) is removed through the forgetting gate to reduce the network burden; secondly, the valid information is filtered from the current sequence input information through the memory gate, and each information component is rated for updating the cell state (Ct-1); and thirdly, the new output (ht) is obtained through the output gate based on the current sequence input and the current sequence state (Ct), which is also the input of the next sequence point [16]. Due to the uncertainty of the optimal offset distance selection, the effective spatial information of the subsurface layer may exist in other positions. The LSTM extracts the optimal offset distance and the information of other adjacent positions simultaneously to improve the robustness of the model.

The attention layer is a network structure that enables a neural network to have the ability to focus on a specific subset of features by automatically learning and calculating the contribution size of the input data to the output data. The attention mechanism can be applied to any complex form of input and can also handle information overload in parallel. The essence of the attention mechanism is to learn the importance of each waveband from the Raman spectral sequence and assign each a weighting factor ($a_{t}$). A many-to-one attention mechanism for Keras was used to build the attention layer [17], where multi-dimensional spatial scattering features ($h_{t}$) share the same attention weights. The output layer is a fully connected network for feature integration and shrimp freshness prediction.

#### 2.4.2. Parameter Setting and Running Environment

Table 1 shows the parameter settings of the attention-based LSTM model. The number of nodes of each LSTM module for scattering feature extraction corresponding to tissue distribution is set to 21. The activation function is set to ‘tanh’ to mitigate gradient disappearance. The number of attention units is set to 50 to extract spectral features corresponding to composition information. Training speed is ensured by reducing the number of model parameters after sample data dimensionality. The number of fully connected nodes is set to 10, and the activation function is ‘ReLU’. The output loss function is set to the mean square error (*MSE*). A gradient descent method is used, while adjusting the appropriate learning rate and training period. The epoch of a single training iteration for all batches is set to 1000. The PyCharm (2021.1.2, JetBrains, Prague, Czech Republic) software and the TensorFlow (2.6.3) package are used for data analysis processing and modeling predictions.

### 2.5. Model Comparison and Evaluation

This research compared the built attention-based LSTM deep learning model with traditional machine learning prediction models combined with an optimal offset distance Raman spectrum. A common predictive modeling method applied in the spectral analysis is partial least squares regression (PLSR). When the number of samples is limited, especially when it is fewer than the number of the characteristics, and the correlation is high, PLSR can be used [18]. Support vector regression (SVR) is a substitute for the traditional method of partial least squares. SVR realizes complex detection tasks through nonlinear kernel function and has better generalization performance [19]. The extreme random tree algorithm (ET) integrates decision trees and a feature point random selection strategy to generate a complex tree network, which has a stronger generalization ability [20].

The training and testing sets were split randomly in a 8:2 ratio. Five-fold cross-validation was used to train the model parameters and avoid model overfitting. The coefficient of determination (*R*^2^), root mean square error (*RMSE*), and residual prediction deviation (*RPD*) were used as model evaluation indicators. The larger the *R*^2^ and *RPD* and the smaller the *RMSE*, the better the fit and prediction of the regression model.

## 3. Result and Discussion

### 3.1. Raman Spectra Analysis

Reference Raman spectra of in-shell shrimp, shrimp shells, and shrimp meat were collected at the laser incidence point by a point laser-based line-scan Raman scattering image acquisition system (i.e., backscatter Raman spectroscopy). Figure 4a shows the normalized Raman spectra of these reference spectra and the typical Raman peaks. The main spectral peaks of shrimp meat are distributed at Raman shifts of 1003, 1148, 1269, 1311, and 1487 cm^−1^. The waveband of 1003 cm^−1^ corresponds to an important amino acid, phenylalanine, which ensures the freshness of shrimp products [21]. The waveband adjacent to 1148 cm corresponds to the C-N stretching vibration of proteins. This waveband serves as the main protein characterization and effectively responds to protein–lipid and protein–protein interactions [22]. The spectral peaks at 1269 and 1311 cm^−1^ correspond to protein secondary structures, which reflects the contribution of C-N stretching and N-H in-plane bending vibrations of the peptide bond as well as the Cα-C stretching and C=O in-plane bending [23]. The Raman peak corresponding to 1487 cm^−1^ characterizes CH_2_ and CH_3_ bending vibrations and CH stretching vibrations. The reduction of the Raman spectral peak at this waveband is influenced by the hydrophobic effect around the aliphatic residues. In contrast to the multiple spectral peaks typically shown by shrimp meat, the main manifestation of shrimp shells is a large fluorescence background. Under the present study, the fluorescence background of shrimp shells may mask the Raman peaks of shrimp shells (e.g., 1148 cm^−1^); meanwhile, the blocking of Raman photons by shrimp shells and the masking of fluorescence also limits the transmission of signals from subsurface shrimp meat, which interferes with the assessment of freshness of shrimp meat. The Raman spectrum of intact in-shell shrimp can also be found to be composed of the spectra of both the shrimp meat and the shrimp shell, but the expression is mainly influenced by the external shell.

As shrimp meat storage time increases and freshness changes, the fat appears oxidized and protein denaturation occurs. The corresponding typical characteristic Raman peaks and fluorescence background also change regularly since the Raman spectrum of shrimp meat at the laser incidence point is somewhat masked by the interference of shrimp shells. Spatially offset Raman spectra were used to demonstrate the Raman spectra and characteristic peak variations of in-shell shrimp with different storage days. As shown in Figure 4b, the Raman peaks at 1148, 1269, and 1311 cm^−1^ corresponding to the secondary structure of the protein decrease gradually with increasing storage time. The increase in the fluorescence background is due to the decomposition of biological tissues and the change and fusion of both the chemical composition and physical distribution as shrimp meat deteriorates. The correspondence between Raman spectral peaks and substances determines that spectral peak identification and intensity can be used to characterize substance composition and content. For an intact in-shell shrimp, which has a complex tissue and composition to be measured, it is difficult to accurately and quantitatively assess the freshness of the shrimp using simple Raman spectral peak correspondence. Conventional modeling and analysis methods generally require first selecting the optimal offset distance Raman spectrum, which is considered to be dominated by the Raman spectral peaks corresponding to the quality of the subsurface shrimp meat. The selected Raman spectrum is combined with a chemometric prediction model to assess the freshness and storage time of the shrimp meat. However, this single offset distance Raman spectrum loses a lot of information on the physical properties of the tissue distribution and the chemical properties of the tissue composition of shrimp meat. Therefore, Raman scattering images consisting of Raman spectra at different offset distances were used as data for modeling and analysis, and the attention-based LSTM model was constructed to further improve the freshness detection accuracy of in-shell shrimp.

### 3.2. Freshness Modeling Prediction Results

The 700 scatter images of 100 shrimps collected and pre-processed within seven days were used as a sample set for the attention-based LSTM model construction, training, and prediction process. Figure 5 shows the box plot of the true values of shrimp storage days and predicted value of attention-based LSTM model. Five independent iterations of the prediction process were performed in the cross-validation with all predicted and true values being stored that were used to produce the resulting statistics in the box plot. The mean of *R*^2^, *RMSE*, and *RPD* predicted by the attention-based LSTM model for the 140 test samples were 0.93, 0.48, and 4.06, respectively. The box plot shows that the mean and median of the predicted deviations are within one day of the true value for the first five days. The attention-based LSTM model can accurately predict the storage date corresponding to freshness when the freshness of shrimp meat changes. According to the interquartile range in the box plot, most of the samples have a concentrated distribution of predicted values in the first five days. The shrimp meat deteriorated on the 6th and 7th day, appearing locally black, oozing tissue mucus, and emitting pungent odor. This also caused sample prediction results to be discrete and outliers to appear. Overall, the model clearly reflects the change in freshness of shrimp meat with the predicted number of storage days.

### 3.3. Validation of Model Structure Rationalization

The network structure of the deep learning model affects the results of freshness assessment of in-shell shrimp. Figure 6 shows the comparison of the network performance of the built attention-based LSTM model and the deleted LSTM layer, attention layer, and fully connected layer in the attention-based LSTM model, respectively. The loss function *MSE* of the training set and the prediction accuracy corresponding to *R*^2^ of the test set are used as two metrics to evaluate the superiority of the network structure in both the training process and the test process. Among them, the *MSE* and *R*^2^ correspond to the one-fold model training process in five-fold cross-validation. After removing the LSTM layer for extracting the physical distribution characteristics and the chemical composition characteristics and removing the attention layer network structure for weighting the output of the LSTM module, the *MSE* of the training process no longer converged after dropping to four. The model under fitting makes it difficult to train an effective shrimp freshness prediction model, while it cannot perform the test set prediction task. The fully connected layer of feature integration has less impact on the convergence speed and performance of the training set but improves the prediction accuracy of the test set. The reason is that the fused features limit the overfitting of the deep model and improve the robustness of the model. Therefore, each layer structure in the attention-based LSTM model was built based on the Raman spectral characteristics of shrimp to jointly achieve the task of shrimp freshness detection.

### 3.4. Model Comparison Results

Compared with the traditional machine learning models, the target-built deep models can extract features more effectively and achieve end-to-end detection tasks. Figure 7 compares the *R*^2^, *RMSE,* and *RPD* of the predictions of the constructed attention-based LSTM deep learning model with those of linear PLSR, nonlinear kernel SVR, and integrated ET models. In the process of training and prediction model by cross-validation, the SVR and ET models performed similarly and both outperformed the PLSR model, which indicates that the Raman spectra are not simply linearly related to the storage date corresponding to freshness. However, although the SVR and ET models have been effective in extracting the optimal offset Raman spectral features, the attention-based LSTM model still has significant advantages with *R*^2^, *RMSE*, and *RPD* values of 0.93, 0.48, and 4.06, respectively. Meanwhile, the smaller prediction standard deviations with *R*^2^, *RMSE*, and *RPD* of 0.005, 0.023, and 0.159 demonstrate the stability of the attention-based LSTM model. This is attributed to the fact that the attention-based LSTM model extracts both chemical composition information and physical distribution information from the Raman scattering images corresponding to the freshness of shrimp meat. The physical information carried by the scattering characteristics of shrimp meat effectively complements the characterization of Raman spectral curves for differences in tissue distribution as freshness changes. Although the structure of the attention-based LSTM model is complex, with sufficient numbers of training samples and iterations, it can achieve the processing of high-dimensional data and high-accuracy detection of complex tasks.

### 3.5. Analysis and Discussion

The translucent characteristics of the shrimp shell ensure that the laser penetrates through the shell to excite the shrimp Raman spectrum, which is transmitted and then accepted through the shell again by the detector. This satisfies the basic requirement for spatially offset Raman spectroscopy; i.e., the subsurface signal can be detected. The subsurface Raman signal is influenced by the thickness and transmittance of the surface layer, and the shelled shrimp selected for this study were intact shrimp with uniform specifications to ensure high-accuracy detection. In the subsurface detection of food with surface interference (e.g., meat products with packaging), the requirements of SORS combined with the attention-based LSTM method proposed in this study can be effectively implemented: (a) Subsurface signals can penetrate surface-interfering substances; (b) The surface interference of the same sample is uniform; (c) There should be no significant difference in surface interference among samples.

The SORS data collected by a line-scan Raman detection device based on a point laser source are complex and high-dimensional information. The conventional SORS detection technique first selects the optimal spatial offset distance, then uses the Raman spectrum detected at that offset position for waveband feature selection, and finally predicts by chemometric model. The technique of choosing the optimal offset distance [10] loses large amounts of physical scattering information. It is difficult for traditional chemometric models to extract effective features and make index predictions for all the Raman scattering images. Therefore, the attention-based LSTM model constructed for the physical distribution and chemical composition of shrimp tissues achieves better analytical results, which also requires sufficient training samples and iterations. The end-to-end depth network model also avoids the artificial error of selecting the offset distance.

The attention-based LSTM model is a deep network structure customized for specific inspection tasks, which can obtain high-precision inspection results under the premise of good consistency of surface layer organization. However, this physical and chemical feature extraction method finds it is difficult to eliminate the interference of surface spectra and cannot adapt to the detection task under the interference of different surface materials. In the deep detection of layered tissues, the surface material exists corresponding to the variation of composition, thickness, optical properties, and other indicators, and the strategy of extracting effective features by considering the Raman hybrid scattering images of the acquired layered samples as a unit is usually vulnerable to the influence of the diverse tissue of the surface layer. Considering the molecular specificity of Raman spectroscopy, the qualitative and quantitative characterization of substance components can be achieved by the position, intensity, and full width at half maxima (FWHM) of Raman spectral peaks, and the blind signal separation technique to separate subsurface Raman spectra without a priori information is an effective method to adapt to different surface interferences.

## 4. Conclusions

A deep learning model of attention-based LSTM is constructed for intact shelled shrimp freshness detection due to the obstruction of shrimp shells and the interference of the shrimp shell’s own Raman signal. The model extracts both physical scattering features and chemical composition features, making it significantly better than the traditional spatially offset Raman selection optimal offset distance analysis method. The reasonability of the structure and performance advantages of the attention-based LSTM model proposed in this study are verified by comparing the prediction results of different structures of the attention-based LSTM model and three other machine learning models. In detection tasks with surface interference, the effective acquisition of deep matter signals is complemented by a rational model structure. Therefore, the selection of a penetrating laser wavelength and power for the material properties and the development of a high-precision detector are also important research elements.

## Figures and Tables

**Figure 1 sensors-23-02827-f001:**
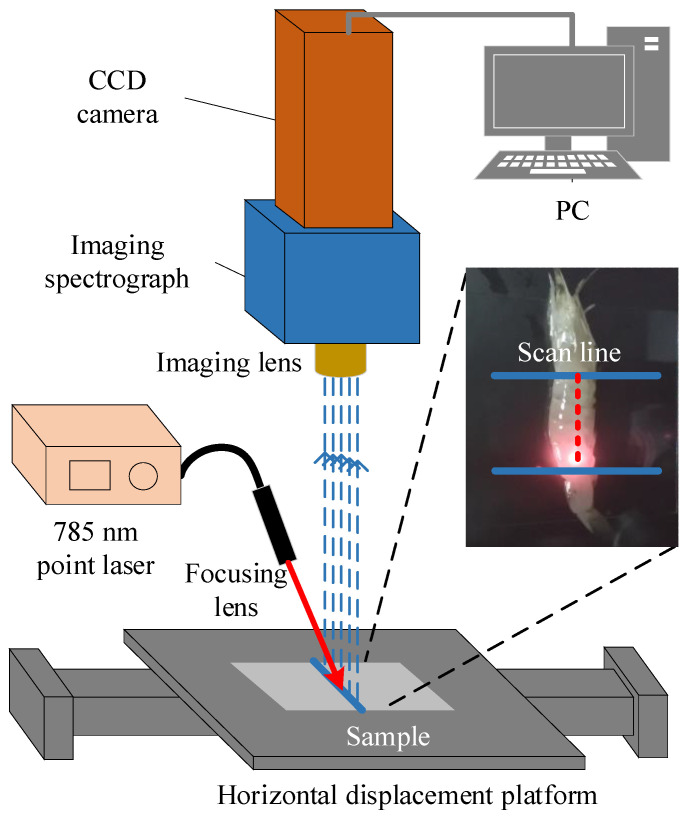
Line-scan Raman scattering image acquisition system.

**Figure 2 sensors-23-02827-f002:**
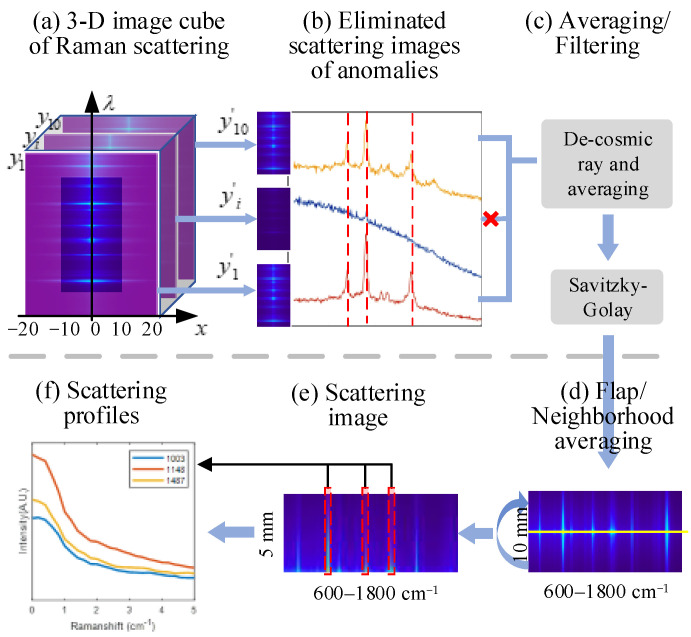
Key steps in the pre-processing of Raman scattering images. (*x* is the spatial position on both sides of the laser spot on each scanning line, *y* is the line-scan image of shrimp at different acquisition positions, and *λ* is the waveband of the Raman spectrum.)

**Figure 3 sensors-23-02827-f003:**
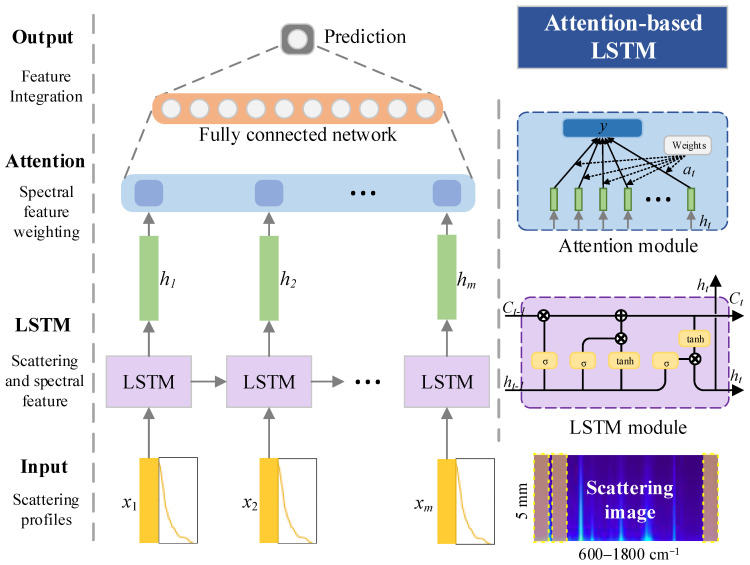
Schematic diagram of the deep network structure of attention-based LSTM. (*m* = 200 is the number of input characteristic bands, *x* is the scattering profiles of different bands, and *h* is the scattering feature extracted by LSTM modules.)

**Figure 4 sensors-23-02827-f004:**
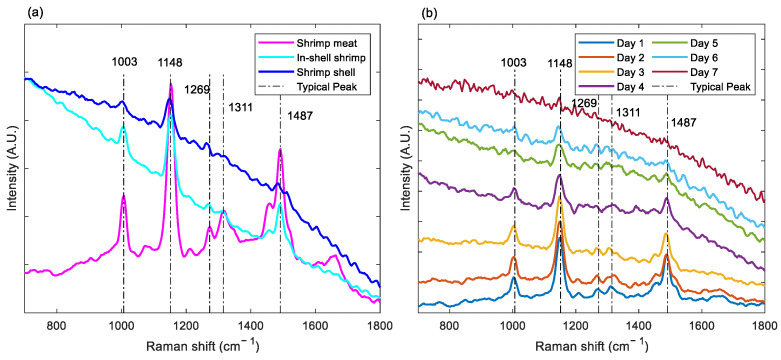
Raman spectra and typical peaks of (**a**) shrimp meat, in-shell shrimp, shrimp shells, and (**b**) spatially offset Raman spectra of in-shell shrimp within seven days of storage.

**Figure 5 sensors-23-02827-f005:**
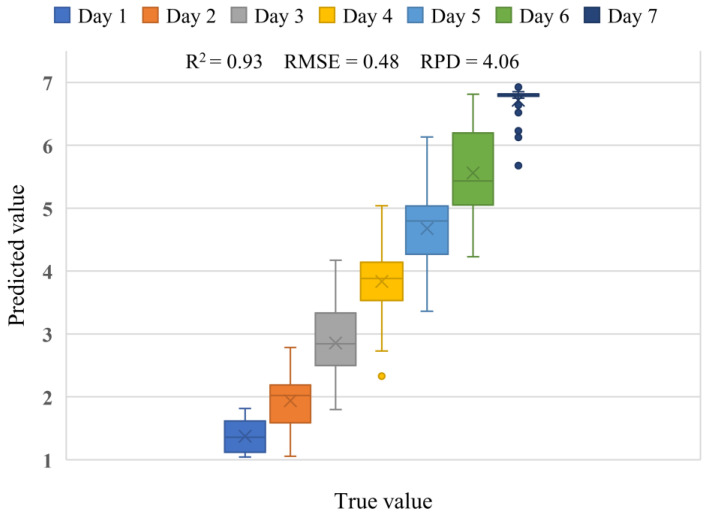
Box plot of the true values of shrimp storage days and predicted value of attention-based LSTM model.

**Figure 6 sensors-23-02827-f006:**
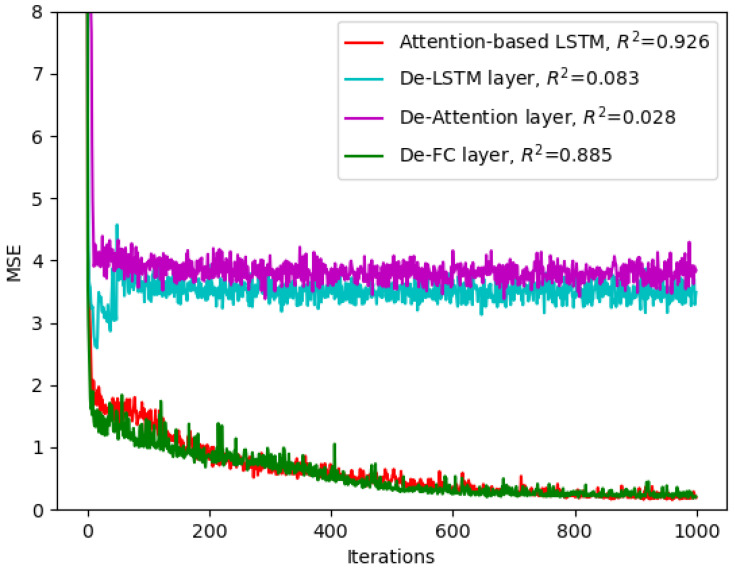
Influence of attention-based LSTM layer structure on loss convergence of training iterative process.

**Figure 7 sensors-23-02827-f007:**
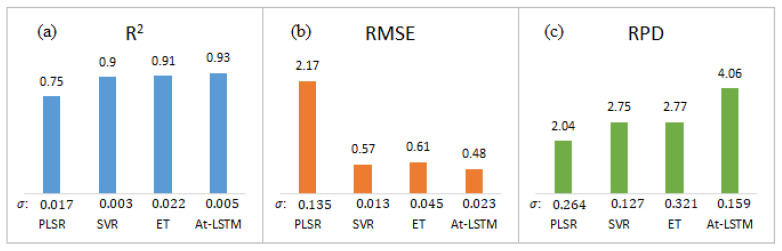
Comparison of evaluation indexes of prediction; (**a**) *R*^2^, (**b**) *RMSE*, and (**c**) *RPD* of PLSR, SVR, ET, and attention-based LSTM model (σ: standard deviation of cross-validation prediction).

**Table 1 sensors-23-02827-t001:** Parameter setting of attention-based LSTM model.

Layer	Input Shape	Units	Activation/Loss	Output Shape	Parameters
LSTM	200 × 11	21	tanh	200 × 21	2772
Attention	200 × 21	50	tanh	50	2541
FC	50	10	ReLU	10	510
Output	10	-	MSE	1	11

## Data Availability

No new data were created or analyzed in this study. Data sharing is not applicable to this article.

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
