# Peer review of "Spatially Offset Raman Spectroscopy Combined with Attention-Based LSTM for Freshness Evaluation of Shrimp"

_sensors, 2023, doi:10.3390/s23052827_

Round 1

Reviewer 1 Report

The authors present interesting results of the application of the deep learning approach to the analysis of Raman spectroscopy data.

Some points should be addressed by the authors to improve the manuscript.

Figure 2 is not clear. 

It is not described what the x and y axes mean. Such a description should be provided in the figure caption and probably in the main text.

In my opinion, the description of data preprocessing in Section 2.3 should be improved. 

Some parts are not clear enough.

The authors mention that they removed some anomalous data but that should be described in more detail.

The authors should improve the caption of Figure 3. The reader should be able to understand the figure without the need to look in the main text for the meaning. For example the meaning of index m, etc.

The authors used a 5-fold cross-validation procedure. That should allow them to obtain not only the mean values of model performance, as presented for example in Figure 7 but also some measure of the dispersion for example standard deviation. The authors should present such results.

In Figure 7 the authors present three measures of model performance. Their meaning is very different so in my opinion presenting them in one figure, using a shared axis is confusing. The authors should split this figure into 3 subfigures for each of the indicators. And as I mentioned earlier display some measure of data dispersion like std dev.

Also, the caption of this future should describe in detail what is presented. The readers should be able to understand the figure without the need to find where it is described in the text.

It is not clear what data are presented in Figure 5. As the authors noticed, they used 5-fold cross-validation. Are these data from one of the folds, or a summary from each repetition of the cross-validation loop?

Reviewer 2 Report

This work is well written. The idea of using Raman Spectroscopy combined with Attention-based LSTM to predict the freshness of shrimp is interesting. Proposed methodology is described comprehensively. Extensive analyses are also performed to validate the findings. I only have some minor comments for authors:

1. Lines 65 - Should be "[11,12]" instead of "[11\12]"

2. Research significance of current work needs to be elaborated.

3. For the sake of fairness, the limitations of current work should be discussed. 

4. Can the same approach be used to predict the freshness of other food products? Please discuss.
